# Reconfigurable optomechanical circulator and directional amplifier

Zhen Shen[1,2], Yan-Lei Zhang[1,2], Yuan Chen[1,2], Fang-Wen Sun[1,2], Xu-Bo Zou[1,2], Guang-Can Guo[1,2] Chang-Ling Zou[1,2] & Chun-Hua Dong[1,2]

Non-reciprocal devices, which allow non-reciprocal signal routing, serve as fundamental elements in photonic and microwave circuits and are crucial in both classical and quantum information processing. The radiation-pressure-induced coupling between light and mechanical motion in travelling-wave resonators has been exploited to break the Lorentz reciprocity, enabling non-reciprocal devices without magnetic materials. Here, we experimentally demonstrate a reconfigurable non-reciprocal device with alternative functions as either a circulator or a directional amplifier via optomechanically induced coherent photon–phonon conversion or gain. The demonstrated device exhibits considerable flexibility and offers exciting opportunities for combining reconfigurability, non-reciprocity and active properties in single photonic devices, which can also be generalized to microwave and acoustic circuits.

---

[1] Key Laboratory of Quantum Information, Chinese Academy of Sciences, University of Science and Technology of China, Hefei 230026, China. [2] Synergetic Innovation Center of Quantum Information and Quantum Physics, University of Science and Technology of China, Anhui 230026, China. These authors contributed equally: Zhen Shen, Yan-Lei Zhang, Yuan Chen. Correspondence and requests for materials should be addressed to F.-W.S. (email: fwsun@ustc.edu.cn) or to C.-L.Z. (email: clzou321@ustc.edu.cn) or to C.-H.D. (email: chunhua@ustc.edu.cn)

The field of classical and quantum information processing with integrated photonics has achieved significant progress during past decades, and numerous optical devices of basic functionality have been realized[1]. Nonetheless, it is still a challenge to obtain devices with non-reciprocal or active gain properties. In particular, non-reciprocal devices, including the common isolator and circulator, have attracted great efforts for both fundamental and practical considerations[2–7]. Although their bulky counterparts play a vital role in daily optics applications, the requirement of a strong external bias magnetic field and magnetic field shields and the compatibility of lossy magneto-optics materials prevent the miniaturization of these devices[8].

Due to the general principle of Lorentz reciprocity or time-reversal symmetry in optics, nonlinear optical effects are one of the remaining options to circumvent these obstacles in the photonic integrated circuit[9–11]. Thus far, optical isolation based on spatiotemporal modulations and three-wave mixing effects have been developed[12–22], and similar mechanisms have been applied to superconducting microwave circuits[23–27]. Yet, very little optimization work has been carried out on optical circulators, another important non-reciprocal device. Circulators, which allow the signal to pass in a unirotational fashion between their ports, can separate opposite signal flows or operate as isolators. For the design of full-duplex systems, circulators are the key elements, offering the opportunity to increase channel capacity and reduce power consumption[28]. Recently, a fibre-integrated optical circulator for single photons was realized, in which non-reciprocal behaviour arises from a chiral interaction between the atom and transversely confined light[29,30]. However, an optical circulator and a directional amplifier for large dynamic range of signal power remain inaccessible.

Here, we demonstrate an optomechanical circulator and directional amplifier in a two-tapered fibre-coupled silica microresonator. Although the non-reciprocal photonic devices based on optomechanical interactions have been demonstrated in previous studies, they are only limited to two-port isolators. The circulator is an indispensable device that cannot be realized by the combination of other optical devices. In contrast, the isolator can be replaced by a circulator by just using two ports of the circulator. In addition, via a simple change in the control field, the device performs as an add–drop filter and can be switched to circulator mode or directional amplifier mode. Our device has several advantages over its bulky counterparts, including reconfigurability, amplification and compactness.

## Results

### Theoretical model.
The optomechanical circulator and directional amplifier feature the photonic structure shown in Fig. 1a, where a silica microsphere resonator is evanescently coupled with two-tapered microfibres (designated α and β) as signal input–output channels. For a passive configuration without a pump, the structure acts as a four-port add–drop filter device[31,32], which can filter the signal from fibre α to β or vice versa via the passive cavity resonance. Because of its travelling-wave nature, the microresonator supports pairs of degenerate clockwise (CW) and counter-clockwise (CCW) whispering-gallery modes, and the device transmission function is symmetric under 1 ↔ 2 and 4 ↔ 3 commutation. The key to the reconfigurable non-reciprocity is the nonlinear optomechanical coupling, represented by the following Hamiltonian:

$$H_{\mathrm{int}} = g_0\left(c_{\mathrm{cw}}^{\dagger}c_{\mathrm{cw}} + c_{\mathrm{ccw}}^{\dagger}c_{\mathrm{ccw}}\right)\left(m + m^{\dagger}\right), \quad (1)$$

where $c_{\mathrm{cw(ccw)}}$ and $m$ denote the bosonic operators of the CW (CCW) optical cavity mode and mechanical mode, respectively. The radial breathing mode can modulate the cavity resonance by

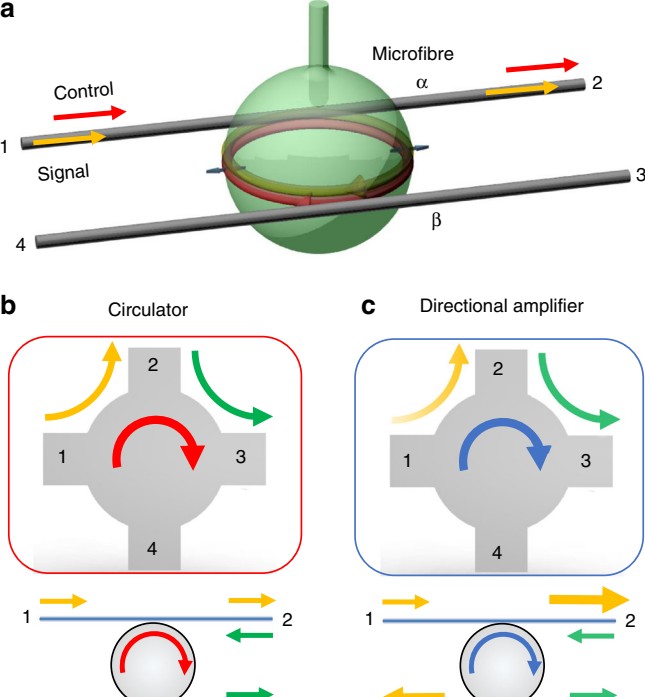

**Fig. 1** Schematic of the optomechanical circulator and directional amplifier. **a** The device consists of an optomechanical resonator and two coupled microfibres. A control field launched into port 1 excites the coupling between the mechanical motion and the clockwise (CW) optical field. **b**, **c** Schematic of the circulator and directional amplifier, respectively. The routing direction of the signal light coincides with the control field (that is, CW direction). For example, the signal entering port 1 is transmitted to the adjacent port 2 as indicated by the yellow arrows, while a signal input at port 2 will drop to port 3 as indicated by the green arrows. The port number follows that of the microfibres in **a**. For the directional amplifier, a larger arrow size indicates a gain in the corresponding direction, while the unchanged arrows represent the absence of gain

changing the circumference of the microsphere, with $g_0$ as the single-photon optomechanical coupling rate.

Biased by a control field that is detuned from the resonance, either the coherent conversion or parametric coupling between the signal photon and phonon can be enhanced[33]. However, the bias control field can only stimulate the interaction between a phonon and a signal photon propagating along the same direction as the bias. As a result of the directional control field, which is chosen as the CW mode in our experiment, the time-reversal symmetry is broken and effective non-reciprocity is produced for the signal light. In particular, the device performs the function of either a circulator or a directional amplifier, which is determined by frequency detuning of the control light with respect to the cavity resonance.

When the CW optical mode is excited via a red-detuned control field, that is, $\omega_{\mathrm{c}} - \omega_{\mathrm{o}} \approx -\omega_{\mathrm{m}}$, where $\omega_{\mathrm{c}}$, $\omega_{\mathrm{o}}$ and $\omega_{\mathrm{m}}$ are the respective frequencies of the control field and optical and mechanical modes, the well-known photon–phonon coherent conversion occurs with a beam-splitter-like interaction $\left(c_{\mathrm{cw}}^{\dagger}m + c_{\mathrm{cw}}m^{\dagger}\right)$[34,35]. For CW signal photons sent to the cavity through fibre port 1(3), as shown in Fig. 1a, a transparent window appears in the transmittance from port 1(3) to port 2(4) when the signal is near resonance with the optical cavity mode. The signal is routed by the control field due to destructive interference between the signal light and mechanically up-converted photons

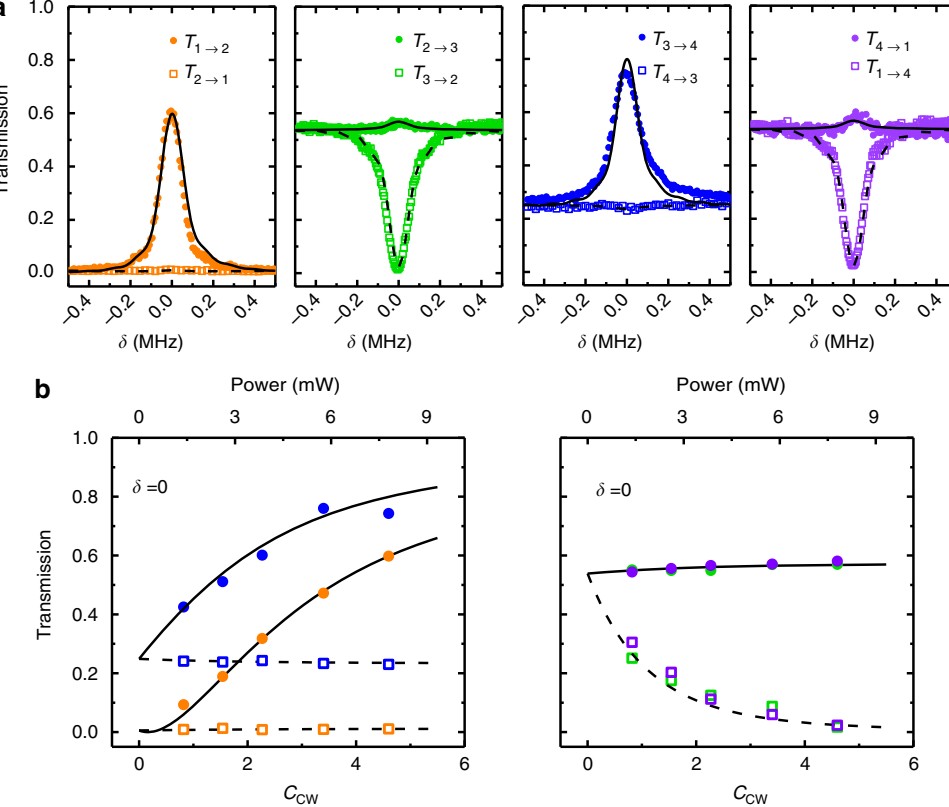

**Fig. 2** Demonstration of the circulator function with a red-detuned control field. **a** Measured port-to-port transmission spectra of the signal around the cavity resonance; the solid circles correspond to $T_{i \to i+1}$ and the open squares correspond to $T_{i+1 \to i}$. The incident control power is 7.8 mW, corresponding to $C_{cw} = 4.6$. **b** The transmittance obtained at $\delta = 0$ versus $C_{cw}$. The lines in **a**, **b** indicate theoretical expectations based on the parameters $\kappa/2\pi = 16.2$ MHz, $\omega_m/2\pi = 90.47$ MHz, and $\gamma/2\pi = 22$ kHz

from the control field[34–36]. In contrast, for the two other input ports 2 and 4, the signal light couples to the CCW optical mode and drops to ports 3 and 1, respectively. Thus, the add–drop functionality is maintained for these two ports in the absence of optomechanical interaction. In general, the device functions as a four-port circulator, in which the signal entering any port is transmitted to the adjacent port in rotation, as shown in Fig. 1b.

For a control field that is blue-detuned from the CW mode ($\omega_c - \omega_o \approx \omega_m$), an effective photon–phonon pair generation process $\left(c_{cw}^\dagger m^\dagger + c_{cw} m\right)$ leads to signal amplification. Similar to the case of a circulator, only a signal launched in a certain direction can couple with the mechanical mode and be amplified, as shown in Fig. 1c. For example, a signal input at port 1 leads to an amplified signal output at both ports 2 and 4. Conversely, a signal input at port 2 will only drop to port 3 without amplification. Therefore, such a device can operate as a common add–drop filter, circulator or directional amplifier by programming the control field.

**Experimental realization.** The optomechanical resonator used in this study is a silica microsphere with a diameter of approximately 35 μm, where we choose a high-Q-factor whispering-galley mode with an intrinsic damping rate $\kappa_0/2\pi = 3$ MHz near 780 nm. The radial breathing mechanical mode has a frequency of $\omega_m/2\pi = 90.47$ MHz and a dissipation rate of $\gamma/2\pi = 22$ kHz (Supplementary Fig. 1). The two microfibres are mounted in two three-dimensional stages and the distance between the resonator and microfibres is fixed throughout the experiment. The external coupling rates of the two channels are $\kappa_\alpha/2\pi = 9$ MHz and $\kappa_\beta/2\pi = 4.2$ MHz, respectively (see Supplementary Note 1 and Supplementary Figs. 2 and 3 for more details regarding the setup).

For an experimental demonstration of the optomechanical circulator, we first measure the signal transmission spectra $T_{i \to i+1}$ from the $i$-th to the $(i + 1)$-th port and the reversal $T_{i+1 \to i}$ for $i \in \{1, 2, 3, 4\}$ (as shown in Fig. 2a) when the CW optical mode is excited by a red-detuned control laser. Here, the control laser and signal light are pulsed (pulse width $\tau = 10$ μs) to avoid thermal instability of the microsphere[19,36]. With the detuning $\delta$ between the signal and the cavity mode (see Supplementary Note 1 and Supplementary Figs. 4–8 for more spectra), the spectra unambiguously present asymmetric transmittance in the forward ($i \to i + 1$) and backward ($i + 1 \to i$) directions around $\delta \approx 0$: relatively high forward transmittance (60–80%) and near-zero backward transmittance. Such performance indicates an optical circulator (Fig. 1b) with an insertion loss of approximately 1–2 dB. The transmission for the backward $T_{4 \to 3}$ is slightly higher because of the imperfection imposed by the unbalanced external coupling rates of the two channels. To obtain a better understanding of the role of the optomechanical interactions, we measure the transmission spectra for control fields of different intensities. The transmissions at $\delta = 0$ are summarized and plotted in Fig. 2b as a function of the cooperativity $C_{cw} \equiv 4g_0^2 N_d/\kappa\gamma$, where $N_d$ is the CW intracavity control photon number, and $\kappa = \kappa_0 + \kappa_\alpha + \kappa_\beta$ is the total cavity damping rate. For increasing $C_{cw}$, we observe that the non-reciprocal transmittance contrast between the forward and backward directions ($T_{i \to i+1} - T_{i+1 \to i}$) increases from 0 to approximately 60%.

By tuning the frequency of the control field to the upper motional sideband of the optical mode ($\omega_c - \omega_o = \omega_m$) and holding the other conditions constant, the same device is reconfigured to act as a directional amplifier. As shown in Fig. 3a, only the signal light launched into port 1 and port 3 (that is,

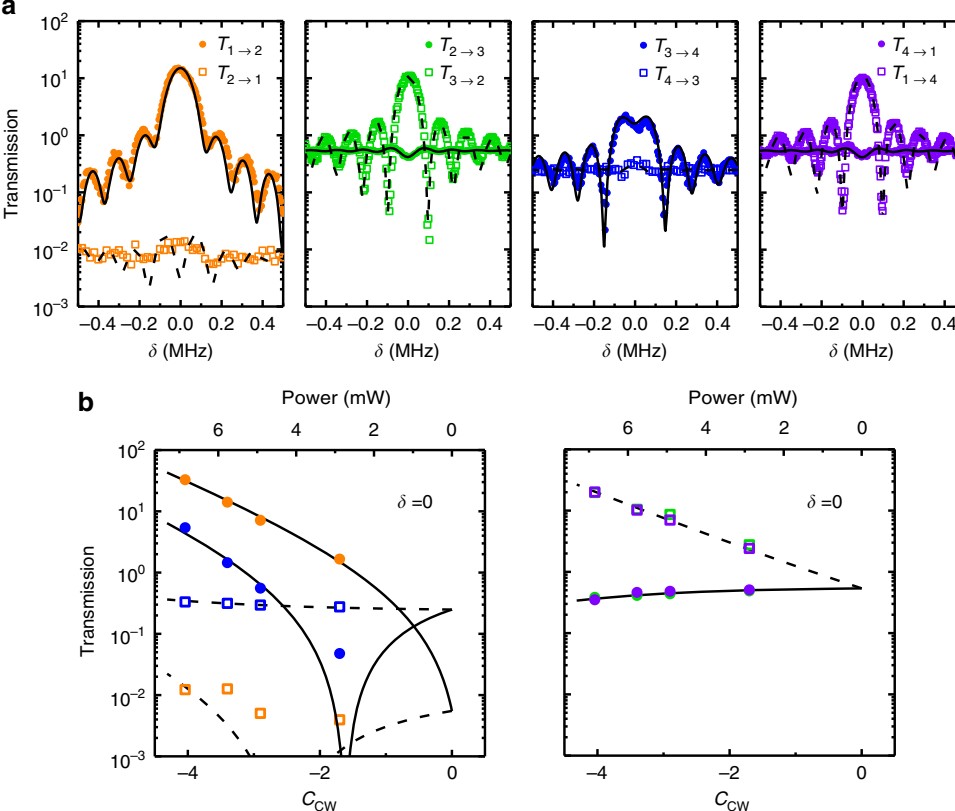

**Fig. 3** Demonstration of a directional amplifier with a blue-detuned control field. **a** Typical measured transmission spectra for the function of the directional amplifier. The solid circles represent $T_{i \to i+1}$ and the open squares represent $T_{i+1 \to i}$. The incident control power is 5.8 mW, corresponding to $C_{cw} = -3.4$. **b** Transmittance obtained at $\delta = 0$ versus $C_{cw}$. The lines in **a**, **b** indicate theoretically expected values based on the parameters $\kappa/2\pi = 16.2$ MHz, $\omega_m/2\pi = 90.47$ MHz, and $\gamma/2\pi = 22$ kHz

coupled to the CW mode) will be simultaneously transferred to port 2 and port 4 with considerable gain, but not vice versa (see Supplementary Note 1 and Supplementary Figs. 4–8 for more spectra). For the channel from port 2 to port 1, the lower transmittance predicted by theory at $\delta = 0$ is not measured due to noise. Here, the experimental results are fitted to the transient transmission spectra, and sinc-function-like oscillations around the central peak are observed due to the impulse response of the device for a 10 μs rectangular control pulse. For the transmittance at $\delta = 0$ and $C_{cw} = -4.0$, where the negative sign of cooperativity represents the blue-detuned drive[19], the signal field from port 1 to port 2 is amplified by 15.2 dB, but in the opposite direction, it suffers a 19.1 dB loss, as shown in Fig. 3b. Hence, the maximum contrast ratio between forward and backward probe transmission is approximately 34.3 dB when $C_{cw} = -4.0$.

## Discussion

To fully characterize the performance of our reconfigurable non-reciprocal devices, we measure the complete transmission spectra $T_{i \to j}$ between all ports (that is, $i, j \in \{1, 2, 3, 4\}$), with $C_{cw} = 0$ for the add–drop filter, $C_{cw} > 0$ for the circulator and $C_{cw} < 0$ for the directional amplifier. Figure 4 shows the experimental results of the transmittance matrix for $C_{cw} = 0, 4.6, -4.0$ at $\delta = 0$, and the matrix for an ideal circulator is given as a comparison (see Supplementary Tables 1 and 2 for the values of all transmission matrices). To quantify the device performance, we introduce the ideality metric $I = 1 - \frac{1}{8} \sum_{i,j} |T_{i \to j}^N - T_{i \to j}^I|$, where $T_{i \to j}^N = T_{i \to j}/\eta_i$ is the normalized experimental transmittance at $\delta = 0$ for

subtracting the influence of the insertion loss (see Supplementary Note 2 for more details), $T_{i \to j}^I$ indicates ideal performance and $\eta_i = \sum_j T_{i \to j}$ is the total output for the signal field entering port $i$. As shown in Fig. 4e, the ideality of the circulator and amplifier approaches unity with increasing $|C_{cw}|$, which agrees well with theoretical fittings (Supplementary Note 2). The best idealities of all three functions of the device exceed 75%. Although the device presented here is a proof-of-principle demonstration, it has provided considerable performance. The mechanism of the device is readily to be realized in the integrated photonic circuits, which is promising for better device performance with higher cooperativity in sophisticated optomechanical structures and collective enhancement in an array of optomechanical cavities[37]. Additionally, the large dynamic range of signal power, potential for low noise, optical real-time reconfigurability and small system size are exciting benefits of this optomechanical approach. Thus, the demonstrated optomechanical reconfigurable non-reciprocal device has great potential for practical applications.

The demonstrated non-reciprocal circulator and amplifier based on the optomechanical interaction in a travelling-wave resonator enable versatile photonic elements and offer the unique advantages of all-optical switching, non-reciprocal routing and amplification. Other promising applications along this direction include non-reciprocal frequency conversion, a narrowband reflector and creation of a synthetic magnetic field for light by exploiting multiple optical modes in a single cavity[19,21,38]. With advances in materials and nanofabrication, these devices can be implemented in photonic integrated circuits[39], which will allow for stronger optomechanical interactions and a smaller device

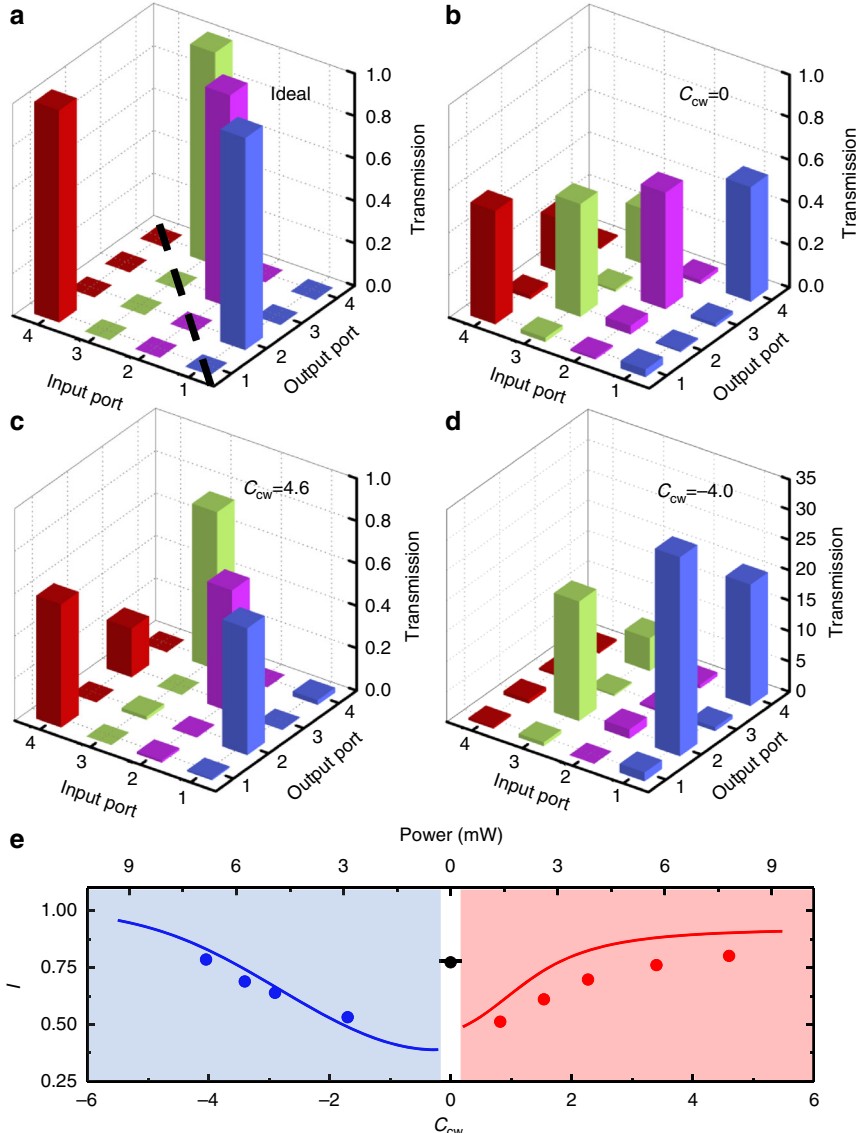

**Fig. 4** Transmission matrices. **a** The transmission matrix of an ideal circulator: $T_{1\to2} = T_{2\to3} = T_{3\to4} = T_{4\to1} = 1$, and all remaining matrix elements are 0. A circulator requires an asymmetric transmission matrix with regard to the dashed line, which breaks the reciprocity. Conversely, **b** shows a symmetric transmission matrix measured without a control field, representing a reciprocal device. **c**, **d** Transmission matrices for the demonstrated circulator and directional amplifier, respectively. The control power is 7.8 mW for the circulator and 6.9 mW for the directional amplifier, corresponding to $C_{cw} = 4.6$ and $-4.0$. The values of all transmission matrices are provided in Supplementary Tables 1 and 2. **e** Identical $I$ of the circulator, directional amplifier and add-drop filter as a function of $C_{cw}$. The lines are the results of theoretical calculations based on the parameters $\kappa/2\pi = 16.2$ MHz, $\omega_m/2\pi = 90.47$ MHz, and $\gamma/2\pi = 22$ kHz. See Supplementary Note 2 for more details of calculation

footprint. Thus, the missing block of non-reciprocity can be tailored and implemented to meet specific experimental demands. The principle demonstrated herein can also be incorporated into microwave superconducting devices as well as acoustic devices in the emerging research field of quantum phononics[40].

During the preparation of this manuscript, a similar work by F. Ruesink et al. has been reported in Nature Communications[41], where an optical circulator based on microtoroid resonator was demonstrated.

**Data availability**. All data generated in this study are available from the corresponding author upon request.

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

## Acknowledgements

The work was supported by the National Key R&D Program of China (Grant Nos. 2016YFA0301303, 2017YFA0304504, 2016YFA0301700), the Strategic Priority Research Program (B) of the Chinese Academy of Sciences (Grant No. XDB01030200), the National Natural Science Foundation of China (Grant Nos. 61575184, 11722436 and 11704370) and the Fundamental Research Funds for the Central Universities. This work was partially carried out at the USTC Center for Micro and Nanoscale Research and Fabrication.

## Author contributions

Z.S., C.-H.D. and C.-L.Z. conceived the experiments. Z.S., Y.C. and C.-H.D. prepared microsphere, built the experimental setup and carried out measurements. Y.-L.Z., Z.S. and Y.C. performed the numerical simulation and analysed the data. X.-B.Z. and F.-W.S. provided theoretical support. Z.S., C.-H.D. and C.-L.Z. wrote the manuscript with input from all co-authors. C.-H.D., F.-W.S. and G.-C.G. supervised the project. All authors contributed extensively to the work presented in this paper.

## Additional information

**Competing interests:** The authors declare no competing interests.

