## [Peer Review File · Nature Communications]

Reviewers' comments:

Reviewer #1 (Remarks to the Author):

This manuscript reports experimental demonstration of reconfigurable optomechanical circulator and directional amplifier in a silica microsphere. In an earlier paper (Ref. 19), the authors have demonstrated optomechanically induced nonreciprocity. In this work, they have extended the earlier work and incorporated the optomechanically induced nonreciprocity in an add-drop filter type of configuration (experimentally, they used two, instead of one, tapered optical fibers for input-output). In addition, they also included the effects of optomechanically induced amplification, for which the control laser is blue-detuned from the cavity resonance (For the optomechanically induced transparency in their earlier work, the control laser is red-detuned from the cavity resonance).

The experimental results presented are of very high quality. The detailed theoretical analysis shows very good agreement between theory and experiment. Overall, the results presented are convincing and conclusive. The question in terms of publication in Nature Communications is whether the manuscript has presented substantially new and significant results. In my opinion, the manuscript has done that. While the key optomechanical processes have been demonstrated in their earlier work, the demonstration of circulator and directional amplifier represents a significant step towards potential applications of these processes. I recommend the publication of the manuscript in Nature Communications, with the following questions and comments for the authors to address:

- 1) The bandwidth of these optomechanical processes is small (a few tens of kHz). Can the authors comment on how to improve the bandwidth or what applications are more suitable for the narrow bandwidth?
- 2) Figure 3 shows sinc-function like oscillations. Why these oscillations do not show up in Fig. 2?
- 3) What are the mode properties or mode numbers of the two closely spaced mechanical modes used in the experiment?
- 4) Figure 1 is somewhat confusing. It took me a while to figure out what the green arrows do. Can the author add a brief description in the figure caption.
- 5) The authors in the abstract emphasized "single photonic structures." I understand single atoms or single optical cavities. What do "single photonic structures" mean and why single?
- 6) The authors stated in the second paragraph of the main text, "nonlinear optical effects are the remaining option to circumvent these obstacles." Do they really mean nonlinear optical effects are the only option, instead of one of the options?

Reviewer #2 (Remarks to the Author):

The paper by Z. Shen et al. describes an experimental realization of a reconfigurable optical circulator and directional amplifier using the radiation pressure interaction in an optomechanical whispering gallery system. The circulator connects the four input-output ports formed by two tapered optical fibers evanescently coupled to the silica microsphere supporting the exploited whispering gallery modes. The protocol is demonstrated with optical pulses and crucially exploits the optomechanical interaction and the application of a control field, which is resonant with the red mechanical sideband of the cavity mode. By shifting the control field to be resonant with the blue sideband of the cavity, the system is changed to a directional amplifier and this is the main reconfigurable aspect of the device.

The device is characterised and its transfer matrix is experimentally determined. The paper is quite clear and the results are interesting because they provide an optical alternative to the recently demonstrated microwave circulators based either on the Josephson or on the electromechanical

interaction. In prospect, the device presented here is promising in terms of on-chip integrability allowing in principle a system which does not need bulky magnetic-based isolators and circulators. On the other hand this is clearly a proof-of-principle demonstration and practical devices of this kind seem to me still far from practical realization.

The paper is interesting and deserves publication in some form, but I am not sure it is relevant and important enough for the broad and generic audience of Nat. Comm. In fact, the proposal is a quite direct extension of published results of the authors' group and of other similar results on nonreciprocal optomechanical devices. More specifically the physical phenomena at the basis of the circulator is optomechanical induced transparency, while the directional amplifier is based on optomechanical induced amplification, which were both demonstrated in Ref. 32 for example and in other papers. The interference between the control field and the applied signal beam is responsible for the destructive interference pathways leading to the circulator behavior. In this respect the device is just a two-fiber extension of a standard whispering gallery mode optomechanical device operated and demonstrated by many groups in the last years.

The present version of the paper does not properly underline the advance with respect to the existing literature and in particular to the nonreciprocal optomechanical devices referenced in the paper (Ref 13 to 27). In conclusion in my opinion the paper is not suitable for publication in Nature Communication.

Response to 1st Referee's comments

General Comments:

This manuscript reports experimental demonstration of reconfigurable optomechanical circulator and directional amplifier in a silica microsphere. In an earlier paper (Ref. 19), the authors have demonstrated optomechanically induced nonreciprocity. In this work, they have extended the earlier work and incorporated the optomechanically induced nonreciprocity in an add-drop filter type of configuration (experimentally, they used two, instead of one, tapered optical fibers for input-output). In addition, they also included the effects of optomechanically induced amplification, for which the control laser is blue-detuned from the cavity resonance (For the optomechanically induced transparency in their earlier work, the control laser is red-detuned from the cavity resonance).

The experimental results presented are of very high quality. The detailed theoretical analysis shows very good agreement between theory and experiment. Overall, the results presented are convincing and conclusive. The question in terms of publication in Nature Communications is whether the manuscript has presented substantially new and significant results. In my opinion, the manuscript has done that. While the key optomechanical processes have been demonstrated in their earlier work, the demonstration of circulator and directional amplifier represents a significant step towards potential applications of these processes.

Reply:

We thank the Referee for the careful reading of our work and the appreciation of the high technical value in our manuscript. In the following, we address the Referee's comments one by one.

Comment 1:

The bandwidth of these optomechanical processes is small (a few tens of kHz). Can the authors comment on how to improve the bandwidth or what applications are more suitable for the narrow bandwidth?

Reply:

Thanks for raising this point. First of all, we answer the approaches to improve the bandwidth:

(1) For the circulator (red-detuning), the bandwidth can be approximated as $(1 + C_{cw})\gamma$ for the interaction time $t \gg (\kappa + \gamma)/[\kappa\gamma(1 + C_{cw})]$ as we did in our experiments. In this case, the bandwidth $(C_{cw} + 1)\gamma = 4N_d g_0^2/\kappa + \gamma$ can be improved by enhance the cooperativity, i.e. increase the optomechanical coupling rate (g_0) by advanced optomechanical microcavity design, increase control photon number (N_d), as well as reduce the linewidth of the optical cavity (κ) while increase the mechanical linewidth γ .

(2) For the directional amplifier (blue-detuning), the transient response effect due to the finite duration of control pulse can't be neglected. For the steady state (longer control pulse), there is a trade-off between the gain and bandwidth, where the amplification diverges for $C_{cw} = -1$. However, for the short control pulse, the system can't reach the steady state even when we have $C_{cw} \leq -1$. Therefore, gain exponentially increases with the pulse duration when the saturation effect can be neglected. In this case, the gain bandwidth is almost independent with the gain, approximately be $|C_{cw} + 1|\gamma = 4N_d g_0^2/\kappa - \gamma$. Therefore, the gain bandwidth can be improved by increasing g_0 and N_d , while reducing κ and γ .

(3) In addition, for both circulator and directional amplifier, the bandwidth can also be improved via an array of optomechanical resonators. For example, in a recent theoretical proposal [arXiv:1707.03339 (2017)], it was theoretically demonstrated that the optomechanical frequency conversion bandwidth can be improved by an array of optomechanical cavities.

For the applications of such circulator and directional amplifier with narrow bandwidth, there may be interesting opportunities for integrated devices for signal filtering and duplexing, source protection for effective noise rejection and sensitive precision measurements.

According to the Referee's comment, we added a section of discussion in the SI to discuss the improvement of bandwidth for the circulator and directional amplifier.

Comment 2:

Figure 3 shows sinc-function like oscillations. Why these oscillations do not show up in Fig. 2?

Reply:

The sinc-function like oscillations in the spectra in Figure 3 is due to the finite optomechanical interaction time for the control pulse used in our experiments. To clarify this, we have provided a section and a figure in the revised supplementary material to explain this.

The intracavity signal field spectra for pulsed control field can be solved, with the asymptotic expression (this expression diverges at $C_{cw} = -1$) as

$$c_{cw}(\delta) = \frac{-\sqrt{\kappa_{in}}\epsilon_{in}}{i\delta - \frac{\kappa}{2}\left(1 + \frac{C_{cw}}{1 - i2\delta/\gamma}\right)} \left[1 + \frac{C_{cw}}{1 - i2\delta/\gamma} e^{i\delta t} A(t)\right]$$

where the transient modification factor $A(t) = 2e^{-(\kappa+\gamma)t/4} \sinh\left[\frac{t\sqrt{(\kappa+\gamma)^2/4 - \kappa\gamma(1+2C_{cw})}}{2}\right]$. In our experiments, we have $\kappa \gg \gamma$ thus

$$A(t) \approx \left(e^{-\frac{(1+C_{cw})\gamma t}{2}} - e^{-\frac{(\kappa-C_{cw})\gamma t}{2}}\right) \approx e^{-\frac{(1+C_{cw})\gamma t}{2}}$$

- (1) For the circulator (red-detuned control), there is no obvious oscillation in the spectrum because the effective linewidth of the mechanical mode is broadened by the optomechanical coupling as $(1 + C_{cw})\gamma t \gg 1$. In this case, the transient factor $A(t) \ll 1$ and thus the transient factor can be neglected and $c_{cw}(\delta)$ can be approximated by the steady-state solution. As a result, there is no oscillation in the measured spectra.
- (2) For the directional amplifier (blue-detuned control), $C_{cw} < 0$ and we even have $C_{cw} < -1$ for pulsed control field. In these cases, $A(t)$ can not be neglected. As a result, there is an oscillation in the spectra due to the interference indicated by the term $1 + \frac{C_{cw}}{1 - i2\delta/\gamma} e^{i\delta t} A(t)$.

The sinc-like oscillations in the spectra have also been observed in the author's previous work and more details can be found in Ref. 19. In the revised manuscript, we have updated the Supplementary Note 2 to include the discussion about the sinc-like oscillation.

Comment 3:

What are the mode properties or mode numbers of the two closely spaced mechanical modes used in the experiment?

Reply:

Similarly to the optical modes, the mechanical modes in a spherical resonator can be identified by radial (q), orbital (l), and azimuth (m) numbers. The two closely spaced mechanical modes used in our experiment are (1, 2, 1) mode and (1, 2, 2) mode. Fig. R1 displays the simulated deformation profile of the two mechanical modes by finite-element method (COMSOL Multiphysics v5.2), with the microsphere diameter used in the simulation is 35 μm according to our experiment. The simulated resonant frequencies are 88.3 MHz and 90.0 MHz, respectively, which are consistent with the experimentally measured frequencies of 87.86 MHz and 90.47 MHz as shown in Fig. S1.

Fig. R1 Simulated deformation profile of (1,2,1) and (1,2,2) mechanical modes.

Comment 4:

Figure 1 is somewhat confusing. It took me a while to figure out what the green arrows do. Can the author add a brief description in the figure caption.

Reply:

Thanks for this nice comment and suggestion. We have added a brief description in the revised figure caption as follows:

“...The routing direction of the signal light coincides with the control field (i.e., CW direction). For example, the signal entering port 1 is transmitted to the adjacent port 2 as indicated by the yellow arrows, while a signal input at port 2 will drop to port 3 as indicated by the green arrows...”

Comment 5:

The authors in the abstract emphasized “single photonic structures.” I understand single atoms or single optical cavities. What do “single photonic structures” mean and why single?

Reply:

According to the Referee’s comment, we changed the statement into “single photonic devices” in the revised manuscript to avoid misunderstanding.

“Single photonic structures” refers to the photonic structure of single silica microresonator coupling with two tapered fibers, which is usually used as an add-drop filter [Ref. 31 & 32 in the manuscript]. Add-drop filters are key elements in the optical add-drop multiplexer. “Add” and “drop” here refer to the capability of the device to add one or more new wavelength channels to an existing multi-wavelength WDM signal [Ref. 31 and Ref. 32]. In our experiment, this photonic structure has new functions of the circulator and directional amplifier via optomechanical interaction.

For a long time, isolators and circulators have been predominantly pursued in photonic structures with magnetized materials, such as based on MZI structure [Ref. 2] or optical resonator [Ref. 8]. On

the other hand, the directional amplifier was proposed and implemented in another optical structure, for example, a photonic-crystal fiber, as a stimulated Brillouin scattering (SBS) waveguide [Ref. 12]. It is still challenging to achieve non-reciprocity and active properties, but our experiment demonstrated the three functions by the same device. Thus, we emphasized that via a simple change of the control field, a single add-drop structure can be switched between add-drop operation mode, circulator operation mode and directional amplifier operation mode.

Comment 6:

The authors stated in the second paragraph of the main text, “nonlinear optical effects are the remaining option to circumvent these obstacles.” Do they really mean nonlinear optical effects are the only option, instead of one of the options?

Reply:

Thanks for pointing out this ambiguous sentence. The nonlinear optical effect is one option to realize non-reciprocity among many different approaches. In the revised manuscript, we changed the statement to: “...*nonlinear optical effects are one of the remaining options to circumvent these obstacles in photonic integrated circuits...*”

Non-reciprocity can also be achieved by other mechanisms beyond a static magnetic field biasing or the nonlinear optics effects. For example, The Sagnac effect can induce a non-reciprocal phase shift for a photon that is propagating in a rotating non-inertial frame of reference. The sound analogue of isolation due to a circulating fluid medium has also been reported [*Science* 343,516–519 (2014)].

Response to 2st Referee's comments

The paper by Z. Shen et al. describes an experimental realization of a reconfigurable optical circulator and directional amplifier using the radiation pressure interaction in an optomechanical whispering gallery system. The circulator connects the four input-output ports formed by two tapered optical fibers evanescently coupled to the silica microsphere supporting the exploited whispering gallery modes. The protocol is demonstrated with optical pulses and crucially exploits the optomechanical interaction and the application of a control field, which is resonant with the red mechanical sideband of the cavity mode. By shifting the control field to be resonant with the blue sideband of the cavity, the system is changed to a directional amplifier and this is the main reconfigurable aspect of the device.

The device is characterised and its transfer matrix is experimentally determined. The paper is quite clear and the results are interesting because they provide an optical alternative to the recently demonstrated microwave circulators based either on the Josephson or on the electromechanical interaction.

Reply:

First of all, we thank the Referee for his or her careful reading of our work and pointing out that the results of this paper are “clear” and “interesting”. In the following, we address the Referee’s comments one by one.

Comment 1:

In prospect, the device presented here is promising in terms of on-chip integrability allowing in principle a system which does not need bulky magnetic-based isolators and circulators. On the other hand this is clearly a proof-of-principle demonstration and practical devices of this kind seem to me still far from practical realization.

Reply:

We thank the Referee for this comment. We agree that the device presented here is a proof-of-principle demonstration of the reconfigurable integrated circulator, add-drop filter and directional amplifier. We are convinced that it is a significant step to impactful applications in practical realizations, although the optomechanical approach for the reconfigurable non-reciprocal devices is still in its infancy. According to the Referee’s concern, here we can take the circulator operation mode as an example to explain the potential for practical realization in more detail:

(1) **Realization in integrated photonic circuits.** As appreciated by the Referee, the demonstrated device is potential for on-chip integration. In the current study, we exploit microsphere (with radius be 17.5 μm) to demonstrate the reconfigurable circulator. The underlying mechanism of our device is actually universal and can be generalized to other optomechanical systems that based on traveling wave optical resonators. There are many reported results on the integrated optomechanics in traveling wave optical resonators, such as the GaAs disk resonator [*E. Gil-Santos, et al., Nature Nanotech.* **10**, 810-816 (2015)], the micro-wheel resonator [*K. Fong, et al., Nano letters* **15**, 6116-6120 (2015)], the racetrack microresonator with under-etching a part of waveguide [*M. Bagheri, et al., Nature Nanotech.* **6**, 726-732 (2011)], as well as the hybrid structures consist of an integrated optical microresonator coupled with a nanomechanical oscillator [*M. Li, et al., PRL* **103**, 223901

(2009)] & [B. Dong, *et al.*, *Micro Electro Mechanical Systems (MEMS), 2015 28th IEEE International Conference on (pp. 49-52). IEEE.*]. Therefore, the demonstrated reconfigurable optomechanical circulator and directional amplifier are readily to be realized in current integrated photonic chips, which allow more compact and robust devices with lower control power.

(2) **The isolation and insertion loss of the device.** In our current demonstration, the isolation of the four-port circulator is (17.4, 15.3, 5.1, 13.4) dB and the corresponding insertion loss is (2.2, 2.4, 1.3, 2.4) dB, for a control power of 7.8 mW, corresponding to $C_{cw} = 4.6$. As explained in detail in the manuscript and Supporting Information, the performance of the device can be improved by increasing the control power, as the coherent interaction between photons and phonon (which is quantified by the cooperativity C_{cw}) is proportional to the intracavity control photon number. Approximately, the imperfections of the device are proportional to $1/(1 + C_{cw})$. In the integrated photonic chip, the optomechanical coupling strength can actually be much higher than that of silica microspheres, due to the smaller sizes and also considerable photon-elastic effects in semiconductors. For example, in microwheel resonator [K. Fong, *et al.*, *Nano letters* 15, 6116-6120 (2015)], the demonstrated cooperativity is around 1 using driving power of approximately 45 μ W. Therefore, we expect that C_{cw} can be further improved by one order based on such integrated microresonator with mW-level control power, and then it is possible to improve the isolation and reduce the insertion loss by more than 10 times.

(3) **The bandwidth of the device.** In our current experimental device, the bandwidth of around 120 kHz can be approximated as $(1 + C_{cw})\gamma$, which can also be improved via the following two approaches:

Approach (i): the bandwidth can be improved to $(C_{cw} + 1)\gamma = 4N_d g_0^2 / \kappa + \gamma$ by enhancing the cooperativity, i.e. increase the optomechanical coupling rate (g_0) by advanced optomechanical microcavity design, increase control photon (N_d) into the cavity mode, as well as reduce the linewidth of the optical cavity (κ) while increase the mechanical linewidth γ . Furthermore, the bandwidth appears to be fundamentally bound by the mechanical frequency, which can be extended to several GHz. For example, the frequency of mechanical breathing mode in GaAs disk resonator is 1.4 GHz [L. Ding, *et al.*, *Appl. Phys. Lett.* 98, 113108 (2011)], the frequency of mechanical thickness mode in AlN microdisk is 10.4 GHz [X. Han, *et al.*, *Appl. Phys. Lett.* 106, 161108 (2015)].

Approach (ii): the bandwidth can also be improved by integrating an array of optomechanical microcavities on taking the advantage of the scalability of the integrated photonic circuits. As proposed in a recent theoretical study [*arXiv:1707.03339 (2017)*], an array of optomechanical cavities can improve the bandwidth of the optomechanical frequency conversion, with the bandwidth proportional to the number of cavities.

In addition, for our device with narrow bandwidth, there may have interesting opportunities for narrowband signal filtering and duplexing, photon source protection for effective noise rejection and sensitive precision measurements.

In summary, our demonstration is readily to be realized in the integrated photonic circuits, which allows the demonstration of an array of compact and robust devices for collectively enhanced interactions, which is also promising for better device performance with higher cooperativity. Additionally, the large dynamic range of signal power, potential for low noise, optical real-time reconfigurability and small system size are exciting benefits of the optomechanical approach to magnet-free nonreciprocal devices. We believe more researchers will join to this exciting new research field, and better devices based on our work will be proposed and demonstrated with more efforts contributed to this field. Thus, the demonstrated optomechanical reconfigurable non-reciprocal device

has great potential for practical applications.

Comment 2:

The paper is interesting and deserves publication in some form, but I am not sure it is relevant and important enough for the broad and generic audience of Nat. Comm. In fact, the proposal is a quite direct extension of published results of the authors' group and of other similar results on nonreciprocal optomechanical devices. More specifically the physical phenomena at the basis of the circulator is optomechanical induced transparency, while the directional amplifier is based on optomechanical induced amplification, which were both demonstrated in Ref. 35 for example and in other papers. The interference between the control field and the applied signal beam is responsible for the destructive interference pathways leading to the circulator behavior. In this respect the device is just a two-fiber extension of a standard whispering gallery mode optomechanical device operated and demonstrated by many groups in the last years.

Reply:

We thank the Referee for this comment. According to this comment, in the revised manuscript we highlighted the main differences and advantages of the device presented here with comparing the previously demonstrated non-reciprocity, which would make the significance of the demonstrated device more clearly.

Since the Referee has the concern about the importance and impact of our work, and the Referee also mentioned the Ref. 35 for the concern about the innovation of our work, we explain the importance, impact, innovation of our work in the following:

- (1) **The importance.** Non-reciprocity devices are indispensable to classical and quantum information processing in photonic integrated circuits. For example, circulators can separate opposite signal flows to increase channel capacity and reduce power consumption for duplex communication systems [Ref. 28]. The well-known to achieve optical non-reciprocity is based on the magneto-optic Faraday effect, which has been commercialized and widely used in practical experiments. However, this approach is unattractive for integrated photonic systems, where small size, integrability and low fabrication cost are important. Furthermore, magneto-optical materials exhibit significant loss at optical frequencies. The device demonstrated here is all-optical and has excellent prospects for on-chip integration. Its reconfigurability offers the advantage of flexibility to realize more complicated signal routing schemes, as it can dynamically switch between different modes of operation. As an example, the switchable directional elements can be the basis of a quantum switch matrix and gain medium, as pursued in atomic quantum information platforms [*Nature Physics* **11**, 37 (2015)].
- (2) **The possible impact.** Our work could be of interest to various research fields, especially integrated photonic, topological photonics, optomechanics and fundamental study on optical non-reciprocity, as the current experiment is an unambiguous demonstration of an effective mechanism for realizing circulator and directional amplifier. Optomechanical non-reciprocity is an emerging research field, where many related experimental and theoretical work have been reported. For example, the optomechanically induced non-reciprocity in travelling wave optical microresonators was proposed in Ref. 18 and demonstrated in our previous work and Ruesink's work [Ref. 19 & 20]. Fang et al. demonstrated non-reciprocity using two coupled optomechanical crystal nanobeam cavities via synthetic magnetism and reservoir engineering

[Ref. 21]. Microwave non-reciprocity is also demonstrated using electromechanical circuits [Ref. 25-27]. Thus, our work could have a great impact on the development of optomechanical non-reciprocity.

(3) **The innovation.** The Referee raises two points, one is about the previous realization of the optomechanically induced transparency (OMIT) and the optomechanically induced amplification (OMIA), and the other one is about the simple structure used in our experiment. Here, we explain the two points in three aspects:

- (i) Our work contains significant innovation, rather than a simple repetition of previous works. The OMIT and OMIA have already been the basic optomechanical effects, and most studies and applications of the optomechanical system are based on the two effects. We also want to stress that the highlight of our work is the realization the magnetic-free non-reciprocal devices, rather than a demonstration of the OMIT and OMIA effects. Although we utilize the OMIT and OMIA effects and appreciate the contribution of Ref. 35, our work has significant new physics.
- (ii) Although the OMIT and OMIA were both demonstrated in Ref. 35, there was no content relative to non-reciprocity proposed in that paper. In fact, the optomechanical crystals used in Ref. 35 were not traveling wave optical microresonators. Our innovation lies in the first proof-of-principle demonstration of a device with alternative functions based on optomechanical interaction. The demonstrated device allows great flexibility with considerable performance.
- (iii) Similar to the case of OMIT/OMIA, the two-fiber-coupled structure is also a very basic photonic structure. A lot of important physical mechanisms or new devices have been demonstrated based on this structure, such as add-drop filter [Ref. 31-32], PT-symmetric system [*Nature Physics* **10**, 394-398 (2014)], frequency comb [*Optica* **1**, 137-144 (2014) & *Optics Express* **24**, 26322-26331 (2016)] and the demonstration of optical chirality [*Phys. Rev. Lett.* **118**, 033901 (2017)]. We need this structure to realize non-reciprocal device, and our work has very different goals and physics. On the other hand, the previously demonstrated optical circulators using two-fiber-coupled structure are based on the photon-atom chiral interaction [Ref. 30] or Kerr effect [*arXiv:1801.09918* (2018)]. Our device has the new function that they do not have, and the non-reciprocity mechanisms are completely different.

In summary, the current work is of significant step that solved the problem on the magnetic-free circulator devices and also demonstrated the directional optical amplifier for the first time. This work may have significant impacts to various research fields, including the integrated photonic circuits, the topological photonics, photon-phonon interactions, and single-photon-level signal processing for both classical and quantum information processing, as well as sensing applications. In addition, our demonstration based on a very simple structure, is readily be realized in an integrated platform (more details are discussed in the Reply to Comment 1), therefore will be attractive to other experimentalists.

Comment 3:

The present version of the paper does not properly underline the advance with respect to the existing literature and in particular to the nonreciprocal optomechanical devices referenced in the paper (Ref 13 to 27). In conclusion in my opinion the paper is not suitable for publication in Nature Communication.

Reply:

We thank the Referee for raising this point. In the revised manuscript, we emphasized the main differences and advantages of our device. Overall, this work is a first proof-of-principle demonstration of the non-magnetic non-reciprocal optical device with alternative functions as either a circulator or a directional amplifier. Although the non-reciprocal photonic device based on optomechanical interactions have been demonstrated, only the simple two-port isolation has been demonstrated in previous studies. The circulator is an indispensable device that can't be realized by a combination of other optical devices. In contrast, the isolator can be replaced by a circulator by just using two ports of the circulator. In addition, the amplifier is also an important device for optical signal processing, but the device has never been demonstrated.

Therefore, our work made a significant contribution to the potential integrated photonics and deserves publishing in Nature Communications, since: (1) **The integrated optical circulator is demonstrated for the first time.** Our demonstrated simplified the previously proposed circulator setup based on the non-reciprocal phase shifter and MZI structure [*Optics Express* **23**, 25118 (2015)]. Our device only requires a single cavity couples to two waveguides, without external phase controlling. (2) **The integrated optical directional amplifier is demonstrated for the first time.** As mentioned by the Referee, the directional amplifier has only been demonstrated in the microwave domain, and there was no optical counterpart.

Here, we discuss the advances of the demonstrated device with respect to the existing literature in the following aspects:

(1) Optomechanically induced non-reciprocity.

Our earlier work [Ref. 19] demonstrated non-reciprocity using optomechanical interactions in a whispering gallery microresonator, as proposed by Hafezi and Rabl [Ref. 18]. Then Ruesink et al. demonstrated isolation in optomechanical resonators [Ref. 20]. Fang et al. demonstrated non-reciprocity in two optomechanical crystal nanobeam cavities [Ref. 21]. **All of the above-mentioned devices are two-port devices, which can't be used as a circulator.** However, our device is a four-port device and realized the circulator. Moreover, in our current study, the transmittance matrix of the device is experimentally determined, which helps to fully characterize the performance of the device. Note that although microwave isolator and circulator has been demonstrated in the electromechanical circuit as we mentioned in the introduction [Ref. 25-27], the optical realization is still have not been reported.

(2) Brillouin-scattering induced non-reciprocity.

Stimulated Brillouin-scattering has been used to achieve opto-acoustic isolators in photonic crystal fiber [Ref. 12]. However, achieving considerable isolation required the photonic crystal fiber to be long, which hindered its integrated application. Then, Brillouin-scattering induced non-reciprocity was demonstrated in whispering gallery microresonators [Ref. 13-14]. In the process of the Brillouin scattering, there are two optical modes and one mechanical mode, which should satisfy the momentum and energy conservation conditions. It is difficult to find two optical modes for the given Brillouin scattering modes, especially in the small size resonators. In contrast, the underlying mechanism of the reconfigurable optomechanical device can be easily generalized in a broad wavelength range.

(3) Non-reciprocity based on three-wave/four-wave mixing effects.

Non-reciprocity based on four-wave mixing and three-wave mixing was demonstrated in silica microresonators [Ref. 16-17] and AlN microresonators [Ref. 15], respectively. Hua et al. demonstrated chip-based isolator via two-fiber-coupled structure with around 18 dB isolation and 3 MHz bandwidth [Ref. 17]. But the two-fiber-coupled structure was used to improve the isolation with parametric amplification instead of realizing a circulator. **Overall, to our knowledge, the**

reconfigurable circulator/directional amplifier has not been demonstrated based on three-wave/four-wave mixing effects.

(4) Non-reciprocity based on spatiotemporal modulations.

Non-reciprocity was demonstrated on-chip [*Phys. Rev. Lett.* **109**, 033901 (2012)] based on interband photonic transitions [Ref. 9]. Limitations in the power of the RF source and loss in the p-i-n junctions led to the weak isolation of 3 dB and large insertion loss of 70 dB. Due to the low dispersion of the waveguide modes, the device exhibited a large bandwidth of 200 GHz. In general, non-reciprocity devices based on waveguide structure have a considerable bandwidth, the ones based on resonance structure (for example the device presented here) exhibit high performance of isolation, insertion loss, smaller size and lower power consumption at the expense of smaller bandwidths, which make them appealing for applications in quantum computing.

(5) Circulator based on Photon-atom interaction.

Recently, a fiber-integrated optical circulator for single photons was realized [Ref. 30]. However, the working wavelength of this circulator is limited to the atomic energy levels and the circulator is valid only for single photon states. Additionally, the photon-atom hybrid system is not robust; people can only realize the single photon circulating by waiting for the single atom flying through the evanescent field.

(6) Kerr non-reciprocity.

A circulator based on Kerr non-reciprocity in microresonators has been demonstrated [*arXiv:1801.09918* (2018)]. This effect can be explained by a Kerr-nonlinearity-mediated interaction between the counter-propagating light waves that induces a splitting of clockwise and counterclockwise resonance frequencies. Despite cavity enhancement, the threshold signal power for non-reciprocity is still high. And its isolation is dependent on the input power (404 mW signal power corresponds to 21 dB isolation), which hinders its application for low power signal. In our experiment, the demonstrated optical circulator and directional amplifier can work for a large dynamic range of signal power.

REVIEWERS' COMMENTS:

Reviewer #1 (Remarks to the Author):

The authors have adequately addressed my questions and comments. I recommend the publication of the manuscript in Nature Communications.

Reviewer #2 (Remarks to the Author):

I have found the new version of the paper and especially the reply to the reviewers report very convincing. The new version now puts the new results in a better context and the relevance of the results is now much better expressed. Therefore I suggest publication in Nature Communication.